# Prevalence and risk factors associated with asymptomatic *Plasmodium falciparum* infection and anemia among pregnant women at the first antenatal care visit: A hospital based cross-sectional study in Kwale County, Kenya

**Gibson Waweru Nyamu** [1,2]*, **Jimmy Hussein Kihara**[3], **Elvis Omondi Oyugi**[4], **Victor Omballa**[5], **Hajara El-Busaidy**[2], **Victor Tunje Jeza**[1]

1 Technical University of Mombasa, Mombasa, Kenya, 2 Department of Health, Kwale County, Kwale County, Kenya, 3 Kenya Medical Research Institute, Nairobi, Kenya, 4 Kenya Field Epidemiology and Laboratory Training Program, Ministry of Health, Nairobi, Kenya, 5 Center for Global Health Research— Kenya Medical Research Institute, Nairobi, Kenya

* wawerugibson2015@gmail.com

## Abstract

### Background

Prevalence of Prevalence of malaria in pregnancy (MiP) in Kenya ranges from 9% to 18%. We estimated the prevalence and factors associated with MiP and anemia in pregnancy (AiP) among asymptomatic women attending antenatal care (ANC) visits.

### Methods

We performed a cross-sectional study among pregnant women attending ANC at Msambweni Hospital, between September 2018 and February 2019. Data was collected and analyzed in Epi Info 7. Descriptive statistics were calculated and we compared MiP and AiP in asymptomatic cases to those without either condition. Adjusted prevalence Odds odds ratios (aPOR) and 95% confidence intervals (CI) were calculated to identify factors associated with asymptomatic MiP and AiP.

### Results

We interviewed 308 study participants; their mean age was 26.6 years (± 5.8 years), mean gestational age was 21.8 weeks (± 6.0 weeks), 173 (56.2%) were in the second trimester of pregnancy, 12.9% (40/308) had MiP and 62.7% had AiP. Women who were aged ≤ 20 years had three times likelihood of developing MiP (aPOR = 3.1 CI: 1.3–7.35) compared to those aged >20 years old. The likelihood of AiP was higher among women with gestational age ≥ 16 weeks (aPOR = 3.9, CI: 1.96–7.75), those with parasitemia (aPOR = 3.3, 95% CI: 1.31–8.18), those in third trimester of pregnancy (aPOR = 2.6, 95% CI:1.40–4.96) and those who reported eating soil as a craving during pregnancy (aPOR = 1.9, 95%CI:1.15–3.29).

**Data Availability Statement:** All relevant data are within the paper.

**Funding:** The author(s) received no specific funding for this work.

**Competing interests:** The authors have declared that no competing interests exist.

## Conclusions

Majority of the women had asymptomatic MiP and AiP. MiP was observed in one tenth of all study participants. Asymptomatic MiP was associated with younger age while AiP was associated with gestational age parasitemia, and soil consumption as a craving during pregnancy.

## Introduction

The commonest plasmodium species that is known to cause malaria in pregnancy (MiP) in Africa is *Plasmodium falciparum* which can lead to anemia in pregnancy (AiP) [1]. The World Health Organization (WHO) in 2019, reported 11 million pregnant women were infected with plasmodium infection in sub-Saharan Africa, resulting in 872 000 low birth weights [2]. In 2019, the Ministry of Health (MOH), Kenya, estimated MiP to be 6.3% among women attending their first antenatal care (ANC) visit [3]. In Kwale County, Kenya, MiP remains a public health concern with a total of 2316 in 2019 [3]. Anemia in Pregnancy is a well-known risk factor for maternal death, stillbirths, low birth weights and infant prematurity [4–6]. Previous studies have reported associations between malaria with AiP [7,8] while consumption of soil (geophagy) has been associated with AiP among African women [9]. Besides plasmodium infections, other known causes of AiP include nutritional deficiencies, infectious diseases like HIV, parasitic infections like hookworm infestation, and hemoglobinopathies [4,9].

MiP constitutes a major risk to the mother, fetus, and neonates including stillbirths, spontaneous abortion, premature delivery, maternal anemia, and low birth weight [2,10]. Due to physiological and immunological changes, pregnant women are more at risk of malaria compared to non-pregnant women living in areas of similar malaria endemicity [11]. Factors associated with MiP are low parity, young age, low maternal education level, early gestational age, young maternal age, fewer previous pregnancies, non-ownership or infrequent use of bed nets and maternal unemployment [7,10,12–14].

ANC is a package given to pregnant women which entails giving prophylaxis and treatment for anemia and malaria among other services, where the AiP is prevented by providing nutritional counseling including iron supplements, and treating cases of AiP [15]. Interventions aimed at prevention and control of MiP adopted by the MOH–Kenya are Intermittent Preventive Treatment (IPTp) of MiP, with sulfadoxine pyrimethamine (SP) given after 12 weeks gestational period done four weeks apart until the pregnant woman delivers. In areas with high malaria transmission such as Western, Nyanza, and Coast regions, Long-Lasting Insecticidal Nets (LLINs) are provided at the ANC during the first contact [16].

Kenya Malaria Indicator Survey (KMIS) 2015 reported two folds increase in the prevalence of malaria in the coastal region compared to KMIS 2010 [17]. In high-transmission regions like coastal regions in Kenya, where levels of acquired immunity tend to be high *P. falciparum* infection is usually asymptomatic in pregnancy. However, parasites may be present in the placenta and contribute to AiP even in the absence of documented peripheral parasitaemia [18]. In the study area, women have been reported delaying in starting pre-natal care during pregnancy which may also contribute to delay interventions measures uptakes like IPTp and LLINs usage hence remain a reservoir of parasites contributing to the spread of the disease from one malaria season to the next [19]. Little information is currently available on the epidemiology of malaria and anemia during pregnancy, in Kwale County except the data collected in passive surveillance according to records in the County.

It is from this background that we carried out this study to estimate the prevalence of asymptomatic MiP and AiP and identify the associated factors among women attending their first ANC visit at the largest referral health facility in Kwale County, Kenya.

## Methods

### Study location

Kwale County is one of six counties in the coastal region of Kenya covering an area of 8,270.3 km$^2$ with a population of 866,820 people [20]. The inhabitants are predominantly Muslim, from the Mijikenda tribe, and practice subsistence farming and smallholder animal husbandry. The weather is hot and humid with two rainy seasons: long rains from April to June and short rains from October through November. The incidence of malaria increases during the long rainy seasons [21].

Msambweni County Referral Hospital (MCRH) is the main referral health facility in Kwale County (Fig 1). The hospital has 155 inpatient beds and189 healthcare workers. Four nurses work at the Mother-Child Health department, who attend on average 125 mothers each month [3].

**Study design and population.** We conducted a cross-sectional study to determine the prevalence of asymptomatic MiP and AiP.

Pregnant women attending their first ANC visit at MCRH between September 2018 and February 2019. Pregnant women seeking their first ANC were included in the study, especially

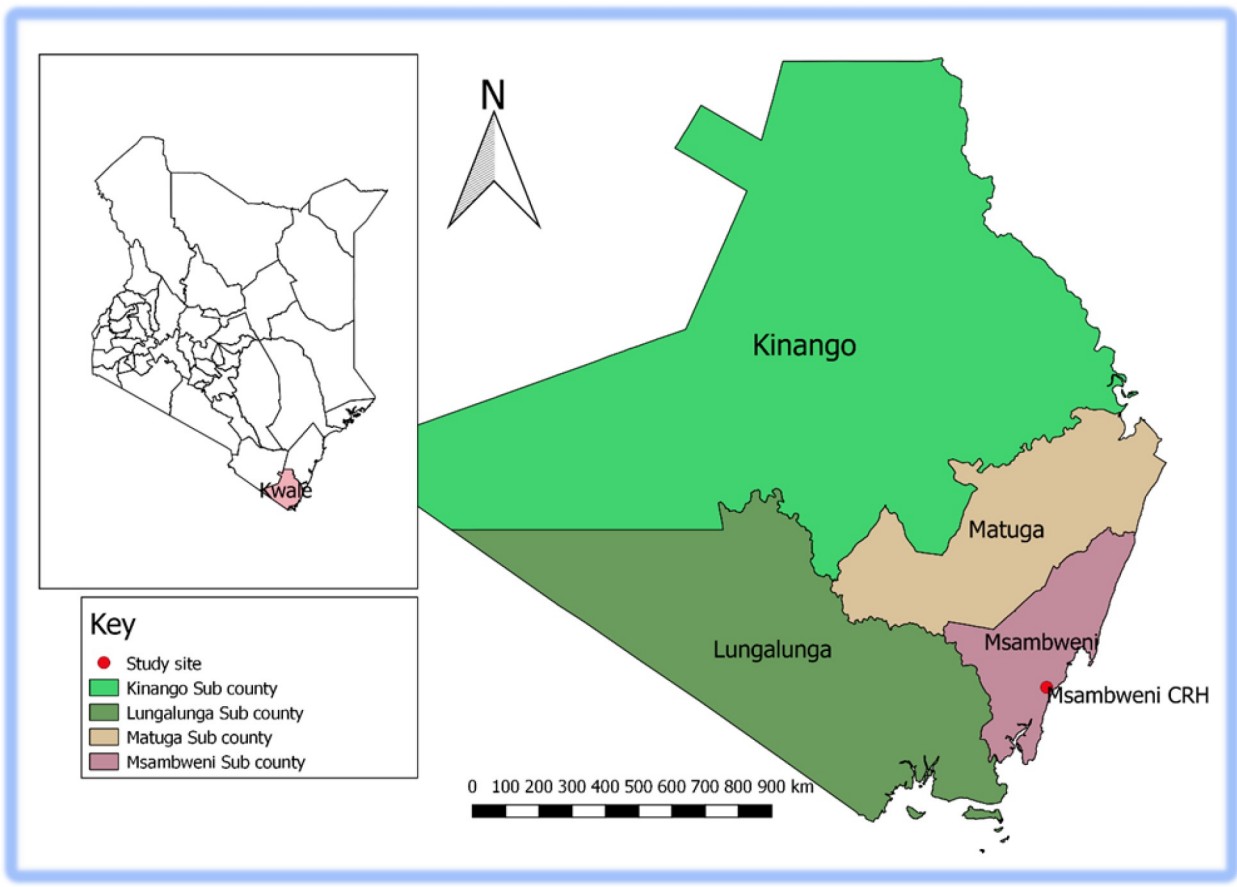

**Fig 1. Map of the study area (Msambweni County Referral Hospital-MCRH).**

those with no symptoms of malaria as per clinical assessment (i.e. no fever (temperature >37.5˚C), chills, rigor, nausea, vomiting, headache, anorexia, or joint/muscle pains).

We excluded pregnant women who had taken anti-malarial drugs within the past two weeks, antipyretics in case they had fever and those receiving micronutrient.

## Definitions of terms

- Asymptomatic malaria was defined as the presence in the peripheral blood of asexual blood-stage of *Plasmodium species*, but has no symptoms of malaria per clinical assessment and the pregnant woman reported not taken antipyretics within 48 hours and antimalarials within 14 days.

- A young age was defined as age ≤ 20 years.

- Anemia was defined as a hemoglobin < 11 g/dl, mild anemia (10–10.9g/dl), moderate anemia (7–9.9g/dl) and severe anemia (<7g/dl).

- First **trimester** was defined as from **week** 1 to the end of **week** 12 while the second **trimester** as from **week** 13 to the end of **week** 26 and the third **trimester** as from **week** 27 to the end of the pregnancy, as it was classified in a study in Ghana [22].

## Sample size determination

Cochran's formula [23] was used to calculate the sample size required to estimate the prevalence of asymptomatic malaria in pregnant women attending their first ANC visit.The study assumed a 95% confidence interval, 80% power, the prevalence of asymptomatic MiP to be 24% [7] and we adjusted by 10% to cater for those who refused to be enrolled. We calculated the desired sample size as 308 participants.

## Sampling procedures

Systematic random sampling method was used to select study participants. Our sampling interval was based on the daily entries in the mother-child health (MCH) register from September 2018 to February 2019. The sampling started by selecting a participant from the daily entries list at random using a table of random numbers and then every $k^{th}$ participant in the frame was selected. A selection interval *(k)* was determined by dividing the total daily entry listed in order to get the number of the participants required per day. If a randomly-selected participant was not eligible for an interview or refused to be part of the study, the next eligible participant on the list was selected. We sampled our study participants until we arrived at our desired sample size of 308 (Fig 2).

Following signing of informed consent, the participants found with malaria parasitemia were treated with Artemether-lumefantrine for those who were in their 2nd and 3rd trimesters while those who were in the 1st trimester were treated with quinine. Those had anemia were given iron supplements and health education on risks and management of anemia during pregnancy at no extra cost.

## Data collection

In-person interviews were conducted using a pre-tested structured questionnaire. The questionnaire was developed in English, and translated to Swahili for non-English speaking respondents.

Variables collected were;

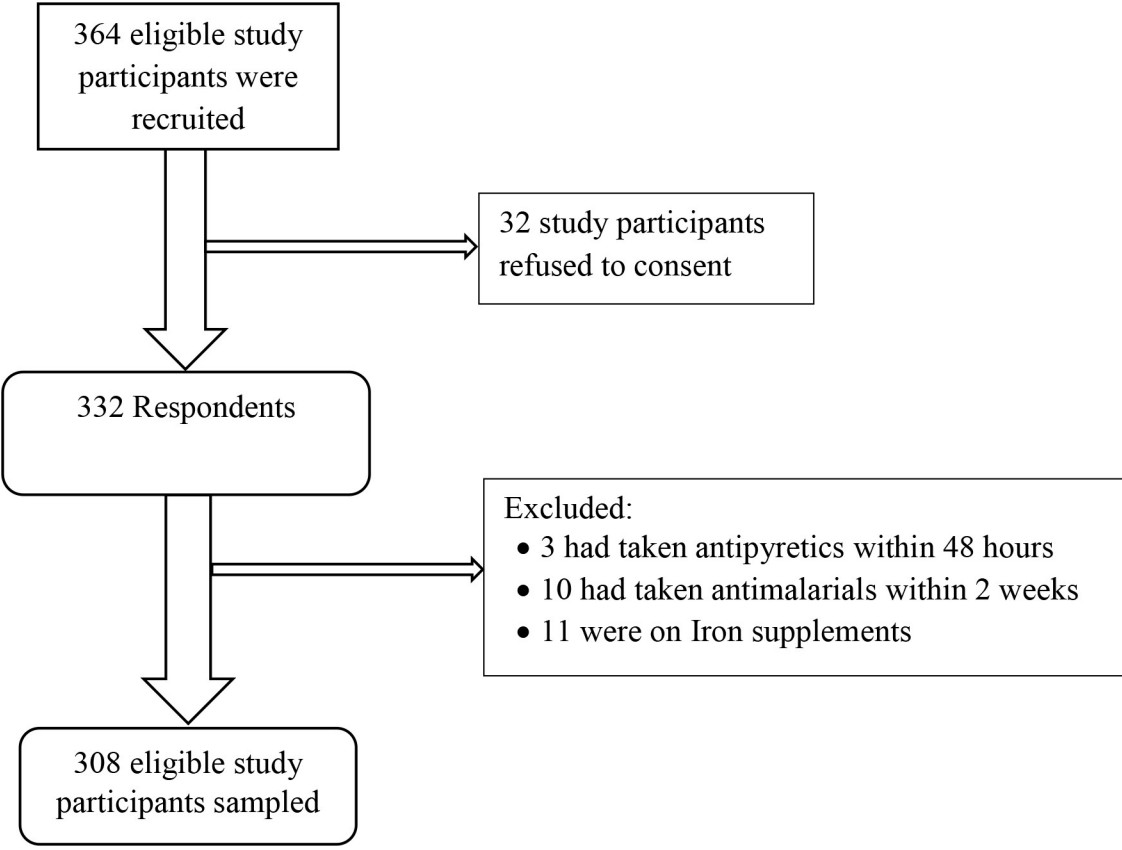

**Fig 2. Flow chart of the recruited participants included in this analysis.**

- **Socio-demographic characteristics** were mother's age, education level, marital status and occupation.

- **Obstetrics variables** were gravidity, parity, trimesters, and gestational age in weeks.

- **Clinical history variables** were history of fever in the last 48 hours and taken anti pyretic drugs, whether the client has taken antimalarial drugs within the last 2 weeks, the tendency of geophagy, and whether or not on iron supplements.

## Laboratory methods

**Hemoglobin testing.** The index finger was cleaned using 70% isopropyl alcohol and pricked using a sterile lancet by well-trained laboratory technicians. The first drop was wiped away using sterile cotton wool, and then the finger was gently squeezed to obtain approximately 30μl drop of blood onto a micro-cuvette and subsequently into a portable heme-analyzer (Hemo Cue Hb 301, Hemo Cue AB 16, Sweden). To determine anemia status, Hb measurement was obtained within 45 seconds and reported in grams per deciliter (gm/dl).

**Malaria testing.** Thick and thin blood films were prepared. Absolute methanol was used to fix thin films and Giemsa stain (3%) staining for 30 minutes. The slides were then rinsed with distilled water and air-dried at room temperature. Slides were then viewed under the microscope using 100x objectives on immersion oil. No Parasite Found was reported after 100 fields were examined and no malaria parasites observed.

Thick films were examined to determine the presence of asexual malaria parasites, quantification of malaria parasites was done by enumerating asexual malaria parasites against 200 white blood cells (WBC). Then, Parasite densities per microliter of blood were determined after multiplying with an assumed WBC count of $8.0^*10^9$/l, with the product of numbers of malaria parasites divided by 200 WBC [24]. Also, the speciation of the *Plasmodium* parasites was done.

**Quality control.** Quality of Giemsa stain was maintained by testing known positives slides. An independent qualified parasitologist examined 10% of both positive and negative slides which were randomly selected and in the case of any disparity they were read by a third parasitologist and his results were deemed final.

Quality control of our hematology analyzer (HemocueHb 301, Hemo Cue AB 16, Sweden) was performed per as the manufacturer instructions, by analyzing dried samples with known Hb levels before testing participant samples.

## Data management and analysis

Data were entered, cleaned, and rechecked using MS Excel 2013. Data were analyzed using Epi Info 7. The following descriptive statistics were calculated: frequencies and proportions for categorical variables, and measures of central tendency (mean, median, and mode) and dispersion (range, interquartile range, and standard deviation) for continuous variables.

We tested the relationship between a variety of predictor variables, including socio-demographic factors and clinical history, and malaria status as the outcome variable, comparing participants who tested malaria positive with those who tested negative. We also compared participants who had anemia to those who did not. Both crude prevalence ratio (cPOR) and adjusted prevalence odds ratio (aPOR), and their 95% confidence intervals were calculated. Variables with $p < 0.05$ were considered statistically significant. Variables with p-values $\leq 0.20$ were included in a logistic regression model using a backward stepwise elimination method to identify independently associated factors.

## Ethics approval and consent to participate

We sought written, informed consent from each participant before interviewing and finger pricking for malaria blood slides and hemoglobin level analysis. The authors used oral consent to accommodate the low literacy rates in the populations served by MCRH and append thumb print where the study participants cannot sign or write on consent form. Permission was granted to conduct the study by Kwale department of health and MCRH director. Ethical clearance was obtained from the Pwani University–Ethical Review Committee (ERC/MSc/021/2018).

## Results

### Socio-demographic characteristics of respondents

A total of 308 respondents were interviewed. Their mean age was 26.6 years (± 5.8 years), 267 (86.9%) were married, 83 (26.9%) had > 8 years of formal schooling, and among them, 29 (9.4%) were formally employed. The mean gestational age was 21.8 weeks (± 6.0 weeks), 29 (9.4%) were in first trimester, 173 (56.2%) were in second trimester and 106 (34.4%) were in third trimester of pregnancy. Those who were primigravidae were 66 (21.4%), secondgravidae were 79 (25.7%) and multigravidae were 163 (52.9%)(Table 1).

Among the participants, 248 (80.5%) owned bed nets (treated or untreated). Of these, 109 (44.3%) had used a bed net for less than 6 months, 34 (13.8%) had used a bed net for 6–12 months, and 103 (41.8%) had used a bed net for > 12 months. In terms of bed net usage in the

**Table 1. Socio demographic characteristics of women attending first antenatal care at Msambweni hospital, Kwale County, Kenya.**

| Characteristics | n (%) |
|---|---|
| **Maternal age (n = 308)** | |
| ≤20 | 50 (16.2) |
| >20 | 258 (83.8) |
| **Education (n = 301)** | |
| Had ≤8 years of formal schooling | 218 (72.4) |
| Had > 8 years Eof formal schooling | 83 (27.6) |
| **Residence (n = 308)** | |
| Rural | 275 (89.3) |
| Urban | 33 (10.7) |
| **Marital status (n = 307)** | |
| Married | 267 (86.9) |
| Single | 32 (10.4) |
| Divorced | 6 (1.9) |
| Widowed | 2 (0.7) |
| **Trimester (n = 308)** | |
| First | 29 (9.4) |
| Second | 173 (56.2) |
| Third | 106 (34.4) |
| **Gravidity (n = 303)** | |
| Primigravidae | 66 (21.8) |
| Secundigravidae | 76 (25.1) |
| Multigravidae | 161 (53.1) |
| **Gestational age in weeks (n = 308)** | |
| <16 | 49 (15.9) |
| ≥16 | 259 (84.1) |
| **Net ownership (n = 308)** | |
| Yes | 248 (80.5) |
| No | 60 (19.5) |
| **Slept under bed net previous night (n = 248)** | |
| Yes | 231 (93.1) |
| No | 17 (6.9) |
| **Frequency of sleeping under bed net (n = 248)** | |
| Always | 205 (82.7) |
| Sometimes | 43 (17.3) |
| **Age of a bed net (in months) (n = 246)** | |
| < 6 months | 109 (44.3) |
| 6–12 months | 34 (13.8) |
| > 12 months3e | 103 (41.9) |

current pregnancy, 231 (93.2%) reported having slept under a bed net the previous night while 205 (82.7%) reported always sleeping under a bed net, and 43 (17.3%) reported sometimes sleeping under a bed net (Table 1).

## Prevalence and factors associated with asymptomatic malaria

Malaria positivity among the 308 study participants was 12.9% (40/308) and the geometric mean parasite count was 3738 parasites per microliter of blood; 35 (87.5%) tested positive for

*Plasmodium falciparum*, 3 (7.5%) *Plasmodium malariae* and 2 (5.0%) *Plasmodium ovale*. In regard to gestational trimesters with plasmodium infections, those in the first trimester were 2/29 (6.9%), second trimester 24/173 (13.9%) and those in the third trimester were 14/106 (13.2%).

The odds of asymptomatic MiP was higher in women who were aged ≤ 20 years (cPOR = 3.5, 95% CI = 1.65–7.23), women who did not own bed nets (cPOR = 2.3, 95% CI 1.08–4.69) and women who owned bed nets but did not sleep under a bed net the night before the interview (cPOR = 2.4, 95% CI 1.14–5.03). After logistic regression analysis, asymptomatic MiP was independently associated with being age ≤ 20 years (aPOR = 4.5 (1.71–12.01) compared with those aged >20 years (Table 2).

## Prevalence and factors associated with anemia

Anemia was reported in 193 (62.7%) participants, and the mean Hb was 9.6 mg/dl (± 1.3mg/ dl); 96 (49.7%) had moderate anemia, 90 (46.6%) had mild anemia and seven (3.6%) had severe

**Table 2. Factors associated asymptomatic malaria parasitaemia in pregnant women, Kwale County, Kenya.**

| Potential factors | N | With Malaria | Crude POR | Adjusted POR |
|---|---|---|---|---|
| | | n (%) | (95% CI) | (95% CI) |
| *Maternal age (n = 308)* | | | | |
| ≤ 20 | 50 | 14 (28) | 3.5 (1.66–7.26) | 4.5 (1.71–12.01) |
| >20 | 258 | 26 (10.1) | 1 | 1 |
| **Education (n = 301)** | | | | |
| Had > 8 years of formal schooling | 83 | 12 (14.5) | 1 | ** |
| Had ≤ 8 years of formal schooling | 218 | 27 (12.4) | 0.8 (0.40–1.74) | ** |
| *Residence (308)* | | | | |
| Urban | 33 | 4 (12.1) | 1 | ** |
| Rural | 275 | 36 (13.1) | 1.1 (0.36–3.29) | ** |
| **Trimester (n = 308)** | | | | |
| Third | 106 | 14 (13.2) | 1 | ** |
| First/second | 202 | 26 (12.9) | 0.97 (0.48–1.95) | ** |
| **Gravidity (n = 303)** | | | | |
| Primigravidae/Secundigravidae | 145 | 23 (15.7) | 1.6 (0.83–3.17) | ** |
| Multigravidae | 163 | 17 (10.4) | 1 | ** |
| **Gestational age in weeks (n = 308)** | | | | |
| <16 weeks | 49 | 7 (14.3) | 1.14 (0.47–2.75) | ** |
| ≥16 weeks | 259 | 33 (12.7) | 1 | ** |
| **Net ownership (n = 308)** | | | | |
| No | 60 | 13 (21.7) | 2.3 (1.09–4.71) | ** |
| Yes | 248 | 27 (10.9) | 1 | ** |
| **Slept under bed net previous night (n = 248)** | | | | |
| No | 56 | 13(23.2) | 2.3(1.14–5.01) | ** |
| Yes | 192 | 22 (11.5) | 1 | ** |
| **Frequency of sleeping under bed net (n = 248)** | | | | |
| Sometimes | 43 | 4 (9.3) | 0.8 (0.28–2.62) | ** |
| Always | 205 | 22 (10.8) | 1 | ** |
| **Age of a bed net (in months) (n = 246)** | | | | |
| ≤12 months | 143 | 12 (8.4) | 1 | ** |
| >12months | 103 | 15 (14.7) | 1.9 (0.83–4.16) | ** |

**CI, confidence interval, N, numbers, POR, prevalence odds ratio, aPOR = adjusted prevalence odds ratio.

**Table 3. Factors associated with anemia in pregnant women, Kwale County, Kenya.**

| Potential factors | N | With anemia | Crude POR | Adjusted POR |
|---|---|---|---|---|
| | | (%) | (95% CI) | (95% CI) |
| **Maternal age (n = 308)** | | | | |
| ≤ 20 | 50 | 34 (68.0) | 1.3 (0.69–2.52) | ** |
| >20 | 258 | 159 (61.6) | 1 | ** |
| *Education level n = (301)* | | | | |
| *Had ≤ 8 years of formal schooling* | 218 | 142 (65.1) | 1.4 (0.85–2.39) | ** |
| *Had > 8 years of formal schooling* | 83 | 47 (56.6) | 1 | ** |
| **Residence (n = 308)** | | | | |
| Urban | 33 | 18 (54.5) | 1 | ** |
| Rural | 275 | 175 (63.6) | 1.5 (0.69–3.03) | ** |
| **Trimester (n = 308)** | | | | |
| First | 29 | 11 (37.9) | 1 | |
| Second | 173 | 102 (58.9) | 2.3 (1.04–5.42) | ** |
| Third | 106 | 80 (75.5) | 5.0 (2.08–12.23) | ** |
| **Gravidity (n = 308)** | | | | |
| Primigravidae | 66 | 42 (63.4) | 1.1 (0.58–1.91) | ** |
| Secundigravidae | 79 | 49 (62.0) | 0.98 (0.56–1.71) | ** |
| Multigravidae | 163 | 102 (62.6) | 1 | |
| **Gestational age in weeks (n = 308)** | | | | |
| < 16 weeks | 49 | 19 (38.8) | 1 | 1 |
| ≥ 16 weeks | 259 | 174 (67.2) | 3.2 (1.72–6.14) | 3.3 (1.72–6.41) |
| **Net ownership n = 308** | | | | |
| No | 60 | 42 (70.0) | 1.5 (0.82–2.80) | ** |
| Yes | 248 | 151 (60.9) | 1 | ** |
| **Eating soil n = 303** | | | | |
| Yes | 117 | 85 (72.7) | 2.1 (1.27–3.45) | 2.0 (1.21–3.41) |
| No | 186 | 104 (55.9) | 1 | 1 |

**CI, confidence interval, POR, prevalence odds ratio, aPOR = adjusted prevalence odds ratio.

anemia. Among those with severe anemia four had malaria, moderate anemia 15/95 (15.8%) and mild anemia were 14/90 (15.6%). Geophagy was reported by 117 (38.6%) participants. Those with a gestational age of ≥ 16 weeks had greater odds of AiP, cPOR = 3.2 (1.72–6.07) compared to those with gestational age <16 weeks. Those who reported eating soil had greater odds of AiP, cPOR = 2.1 (1.27–3.45) compared with those who did not report eating soil. Following logistic regression analysis, AiP was independently associated with gestational age ≥ 16 weeks (aPOR; 3.3, 95% CI: 1.72–6.41), and those who reported eating soil (aPOR 2.0, 95% CI: 1.21–3.41) Table 3.

Pregnant women with malaria parasitemia were three times more likely to have anemia compared to those without malaria parasitemia ($\chi^2$ = 2.79, P-value = 0.005, (aPOR; 3.5, 95% CI: 1.21–8.60) (Table 4).

## Discussion

Numerous studies have shown that anemia and malaria contribute to morbidity and mortality among pregnant women. In this study, we found that one in eight women had asymptomatic MiP and more than half of the women had AiP. Asymptomatic MiP was associated with young age (≤ 20 years). Anemia prevalence was also associated with pregnant women who

**Table 4. Chi-square analysis of proportions with and without malaria parasitemia and anemia.**

| Malaria parasitemia | Anemia | | Total | Crude POR | $\chi^2$ | p value | Adjusted POR |
|---|---|---|---|---|---|---|---|
| | Yes | No | | | | | |
| Yes | 33 (17.1%) | 7 (6.1%) | 40 | 3.2 (1.36–7.46) | 6.8 | 0.009 | 3.5 (1.46–8.60) |
| No | 160(82.9%) | 108 (93.9%) | 268 | ref | | | ref |
| Total | 193 | 115 | 308 | | | | |

POR, prevalence odds Ratio, Ref, reference.

reported eating soil, were in their first and third trimesters of pregnancy, and had *P. falciparum* infections.

The prevalence of asymptomatic MiP in our study population was 12.9%, which is similar to the prevalence among pregnant women in Ethiopia (9.1%) [25] but lower than the prevalence found in Burkina Faso (24%) among pregnant women [7]. In this study, we included only pregnant women seeking ANC services for the first time in their current pregnancy and the majority of study participants (82.7%) reported always using bednet, whereas the study in Burkina Faso included pregnant women seeking ANC services at any point in their pregnancies. These may be factors that lead in the difference between this two reported prevalence. In this study, we found that the odds of MiP among young women ($\leq$ 20 years of age) were greater than the odds of MiP among older women. This may reflect continuing development of malarial immunity [13,26,27]. Contrast to our findings, a study in Gabon showed that there is no difference found between younger and older pregnant women [28].

The current study reports higher proportion of *P. falciparum* infections in pregnant women who were in second and third trimesters and less proportion in the first trimester. Studies in Nigeria have also reported high malaria prevalence in pregnant women who were in their second trimester [29,30]. Another study in Mali reported pregnant women in their first trimester were two times more likely to get malaria compared to the third trimester [31]. The probable explanation may be our study had a few numbers in the first trimester compared with 2$^{nd}$ and 3$^{rd}$ trimesters hence the higher prevalence. With an increase in the number of pregnant women in their first trimester, there is the possibility that there could be changes from the present results.

In this study, we reported the highest proportion of Plasmodium infections among the primigravidae (19.7%) followed by secundigravidae (12.7%) and multigravidae (10.4%) with parasitemia declining with increasing gravidity.

These results are consistent with previous studies which found *Plasmodium* infections are more common in primigravidae compared to multigravidae [7,30,32]. The reason for the present result of gravidae-associated predisposition to *P. falciparum* infections may be due to the fact that adults who live in malaria-endemic regions generally have some acquired immunity to malaria infection. This acquired immunity diminishes significantly in pregnancy particularly in primigravidae. It has also been suggested by various authors that the early onset of antibody response in multigravidae and the delayed antibody production in primigravidae may be responsible for the gravidity-dependent and differential prevalence of *falciparum* malaria among pregnant women [18,33].

Intervention measures for first visit pregnant women at antenatal clinic for malaria in pregnancy are IPTp and provision of LLINs among others [16]. Although the role of IPTp is known to reduce maternal malaria episodes and improve pregnancy outcomes [34], the current study did not include pregnant women who had taken IPTp hence we could not ascertain the role played by IPTp. Our study evaluated through interviewing study participants on bed net ownership and usage. Majority of our respondents owned, slept under a bed net and

almost all participants reported sleeping under a bed net the previous night. The KMIS, 2015, reported similar high rates of bed net ownership and use in pregnant women living in malaria-endemic zones, with 83.7% of pregnant women reporting sleeping under a bed net the night before they were interviewed [17]. Bed net ownership and usage have been reported in several studies to be protective against malaria infections [27,35]. Our observed high rate of bed net ownership may be the result of Kenya's Malaria Control Program conducted a mass net distribution campaign and all households were issued bed nets in Kwale County, in 2017 [36]. Bed nets ownership and usage have been reported to have protective effects for *Plasmodium* infections [37]. In this study, pregnant women with no bed nets had 90% higher odds of asymptomatic MiP compared to those who owned bed nets, though was not statistically significant.

The study reported more than half of the respondents (60%) had AiP. The etiology of anemia is variable and potentially multi-factorial, and thus several underlying morbid and co-morbid conditions may contribute to the prevalence of anemia. Similar to our findings, one study conducted in the Pumwani maternity hospital in Nairobi, Kenya reported an AiP prevalence of 57% [38], and another study in southern Ethiopia reported an AiP prevalence of 60% [34]. Lower prevalence of AiP have also been reported in southwest Ethiopia (23.5%) and northwest Ethiopia (16.5%) [39,40].

The present study found that a significant number of women with asymptomatic MiP had AiP. Malaria in pregnancy is known to cause AiP, this association has also been reported in other studies in sub-Saharan Africa [7,8].

We also found high reported rates of eating soil (geophagy), consistent with other studies that have found high rates of geophagy among pregnant African women as well as associations between geophagy and anemia [9,41].

Our results indicate that, anemia is more common among women in their third trimester than women in their first trimester, similar to findings reported in other studies [42,43]. Hemoglobin decreases through to the end of the third trimester. Anemia is a function of plasma volume and red cell mass; both of which increase during pregnancy; but the increase in plasma volume is proportionately greater than the increase in red cell mass [44].

## Our study had several limitations

We collected data from September 2018 to February 2019, a period during which there is low malaria transmission. This could have resulted in the under estimation of the overall prevalence of asymptomatic MiP in our study area. A continuous monitoring throughout the year of MiP incidence will account for seasonality burden [21].

In addition, the study was hospital-based, excluding pregnant women who did not seek ANC services. While this may limit the generalizability of findings to the community, few women fail to seek antenatal care in our study area. Determination of factors associated with asymptomatic MiP and AiP in hospital-based studies provides a proxy indicator of predictors in the community of that particular facility when community-based surveys are not feasible.

Lastly, this study did not explore other factors that may contribute to anemia, including nutritional factors, soil-transmitted helminthes infection, and hereditary conditions such as sickle cell disease thus limiting our ability to assess the contribution of other causes of anemia during pregnancy. However, diagnosis of anemia was based on laboratory analysis and did not depend on clinical assessment as reported by other researchers.

## Conclusion

Asymptomatic *Plasmodium* infections and anemia are common in women attending their first ANC visit at Msambweni County Referral Hospital in Kwale County. Most of the *Plasmodium*

infections in this area are caused by *P. falciparum*. We did not observe a clear gravidity pattern of asymptomatic MiP, however was associated with younger maternal age (≤20 years). Anemia in pregnancy was associated with *Plasmodium* infections, women who reported to have geophagy tendency and those who were their third trimester. Majority of the study participants in this study registered for antenatal care in their second and third trimester. This practice is detrimental as it does not allow for early detection and correction of pregnancy related complications such as anemia. Therefore, we recommend to the Msambweni County referral hospital in conjunction with Kwale department of health should organize for regular outreach to the community targeting pregnant women for health education should be provided to positively influence the knowledge and attitudes of pregnant women of child-bearing age to malaria and anemia, including early antenatal registration.

## Acknowledgments

We would like to thank the Technical University of Mombasa, Kenya, for their support during the study. Special thanks go to study participants and to the Kwale County Department of Health authorities for their collaboration. We also thank the many Vector Borne Disease Control Unit (VBDCU) Laboratory officers, including Said Lipi, Joyce Bandika, Peter Siema, Charles Ng'ang'a, and Elton Mzungu for their dedication and meticulous microscopy. We extend a special thank you to nurse Wendy Losier, for her assistance with the consenting process and data collection. We also thank Dr. Robert Perry, Dr. Elizabeth Wanja, and Dr. Shama Cash-Goldwasser for their valuable insight and comments on the manuscript.

## Author Contributions

**Conceptualization:** Gibson Waweru Nyamu, Victor Tunje Jeza.

**Data curation:** Gibson Waweru Nyamu, Victor Omballa.

**Formal analysis:** Gibson Waweru Nyamu.

**Funding acquisition:** Gibson Waweru Nyamu.

**Investigation:** Gibson Waweru Nyamu, Hajara El-Busaidy.

**Methodology:** Gibson Waweru Nyamu, Jimmy Hussein Kihara, Elvis Omondi Oyugi, Victor Omballa.

**Supervision:** Jimmy Hussein Kihara, Elvis Omondi Oyugi, Hajara El-Busaidy, Victor Tunje Jeza.

**Validation:** Gibson Waweru Nyamu, Jimmy Hussein Kihara, Victor Omballa, Hajara El-Busaidy, Victor Tunje Jeza.

**Visualization:** Gibson Waweru Nyamu, Jimmy Hussein Kihara, Elvis Omondi Oyugi, Victor Omballa, Hajara El-Busaidy, Victor Tunje Jeza.

**Writing – original draft:** Gibson Waweru Nyamu.

**Writing – review & editing:** Gibson Waweru Nyamu, Jimmy Hussein Kihara, Elvis Omondi Oyugi, Victor Omballa, Hajara El-Busaidy, Victor Tunje Jeza.

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
