## [Decision Letter · Decision Letter 0]

14 May 2020

PONE-D-20-11384

Prevalence and factors associated with asymptomatic malaria and anemia among pregnant women in Kwale County, Kenya: A hospital based cross sectional study

PLOS ONE

Dear Mr Nyamu,

Thank you for submitting your manuscript to PLOS ONE. After careful consideration, we feel that it has merit but does not fully meet PLOS ONE’s publication criteria as it currently stands. Therefore, we invite you to submit a revised version of the manuscript that addresses the points raised during the review process.

Both the expert reviewers have expressed significant concerns in several areas, many of which are overlapping, and with all of which I entirely agree. Your revised manuscript, should you choose to submit such, must therefore take into account all their specific comments and suggestions for improvement. The statistical analyses especially need attention as indicated in the detailed comments of Reviewer #2 in particular. There are also typographical/grammatical errors throughout that require your close attention.

We would appreciate receiving your revised manuscript by Jun 28 2020 11:59PM. To enhance the reproducibility of your results, we recommend that if applicable you deposit your laboratory protocols in protocols.io, where a protocol can be assigned its own identifier (DOI) such that it can be cited independently in the future. For instructions see: http://journals.plos.org/plosone/s/submission-guidelines#loc-laboratory-protocols

We look forward to receiving your revised manuscript.

Kind regards,

Adrian J.F. Luty, PhD

Academic Editor

PLOS ONE

2. Please carefully proofread your manuscript for typographical errors. For example, in the abstract “Prevalence of malaria in pregnant (MIP) in Kenya …” and in the methods section “We conducted across-sectional study …”.

4. Please provide details of the obtained Ethics approval and the informed written participant consent in the methods section of your manuscript. Currently this information is only available in the ethics statement on the online submission form.

6. Please amend either the abstract on the online submission form (via Edit Submission) or the abstract in the manuscript so that they are identical.

7. We note that Figure 1 in your submission contain map images which may be copyrighted. All PLOS content is published under the Creative Commons Attribution License (CC BY 4.0), which means that the manuscript, images, and Supporting Information files will be freely available online, and any third party is permitted to access, download, copy, distribute, and use these materials in any way, even commercially, with proper attribution. For these reasons, we cannot publish previously copyrighted maps or satellite images created using proprietary data, such as Google software (Google Maps, Street View, and Earth). For more information, see our copyright guidelines: http://journals.plos.org/plosone/s/licenses-and-copyright.

8. Please include your tables as part of your main manuscript and remove the individual files. Please note that supplementary tables (should remain/ be uploaded) as separate "supporting information" files

9. We note you have included a table to which you do not refer in the text of your manuscript. Please ensure that you refer to Table 3 in your text; if accepted, production will need this reference to link the reader to the Table.

Reviewers' comments:

Reviewer's Responses to Questions

**Comments to the Author**

1. Is the manuscript technically sound, and do the data support the conclusions?

Reviewer #1: Partly

Reviewer #2: No

2. Has the statistical analysis been performed appropriately and rigorously? 

Reviewer #1: Yes

Reviewer #2: No

3. Have the authors made all data underlying the findings in their manuscript fully available?

Reviewer #1: Yes

Reviewer #2: Yes

4. Is the manuscript presented in an intelligible fashion and written in standard English?

Reviewer #1: Yes

Reviewer #2: No

5. Review Comments to the Author

Reviewer #1: The investigators have surveyed pregnant women in their district of Kenya for malaria and anaemia at first antenatal visit. With a cohort of just over 300 women they have a reasonable power to identify risk factors for these conditions, as well as estimating their prevalence with reasonable precision. The observations are not particularly novel, but the study is of some interest in examining these important pregnancy-associated morbidities.

Major comments

1. Abstract results: make it clear whether the results presented are adjusted odds ratios throughout.

2. Risk factors for AIP: The authors need to better explain how anaemia risk is apparently associated with BOTH first trimester, AND gestation >16 weeks. There are problems with results presentation in Table 3 and it seems they have got the reference and comparator groups switched for the gestation comparison. In addition, part of the results in the table are presented as percentages of women with risk factor who are anaemia (the table column heading) and part are presented as percentage of anaemic women who fall into each risk factor category (not what we want). Please redo this table and recheck your AORs and revise text accordingly.

3. Discussion: differences in malaria prevalence between Kenya and Burkina Faso are more likely related to malaria transmission intensity, prevalence falls with gestation, and falls more in women receiving IPTp. (Related to this, please also discuss role of IPTp in controlling MIP in conclusions).

4. Similarly, end P 11, haemodilution over pregnancy seems more likely to explain declining hb over pregnancy. If postulating dfetal needs please provide reference.

Minor comments

1. Abstract: line 1 “pregnancy”. Results 3rd sentence rewrite “ Women who had MIP were 12.9%”. Last line of results: rewrite for clarity.

2. There were a lot of spacing issues in the pdf, e.g. words joined together, or words and brackets without spaces before them. Examples in abstract methods: odds ratios(OR) and confidence intervals(CI)- there are many others.

3. Introduction 3rd sentence change “malaria with AIP and consumption of soil (geophagy)”, needs different punctuation?

4. Next sentence: is this risk factors for MIP (parasitic infections? High gravidity is NOT associated with MIP) Or AIP?

5. “In sub-Saharan Africa, MIP affects approximately 125 million pregnant women every year”- this is a major misreading of the cited reference.

6. Top of P 5 “a cross sectional”

7. Inclusion/exclusion: “as well as those who”

8. Sample size “desired level” of what was 5%?

9. Data management “ between a variety…”

10. Page 8: “prevalence and risk factors for malaria parasitemia”. Latent malaria is not a recognised phrase. Next line remove “who”.

11. Parasite counts are best expressed as geometric mean not median.

12. P 9 define mild, moderate, severe anaemia.

13. Page 11 first paragraph contains multiple grammatical errors, please rewrite

14. Third paragraph same page please rewrite.

15. A lot of the references are incomplete (lacking volumes/pages), or not enough details are given to retrieve them, or have formatting issues. Refs 2, 5, 6 (what is MOH? Where can this be obtained- similar comments for some others), 7, 17-22, 33.

16. Table groups need editing to ensure e.g. it is clear which group women with 8 y school fall into. Same for gestation. And make table 2 N column wider so numbers are readable.

17.

Reviewer #2: Malaria in pregnancy (MiP) remains a serious public health issue in sub-Saharan African countries despite efficient preventive measures, making difficult for the overall control of malaria. This present study aims to determine the prevalence and risk factors associated with asymptomatic malaria and anaemia among pregnant women in Kwale county, Kenya. The topic of asymptomatic malaria (potential reservoir) is relevant, however, as presented, this article is not original. In addition, the manuscript still requires significant grammatical editing to improve its readability and clarity.

The authors should highlight the fact that asymptomatic malaria is an important concern especially in the first ANC visit where any preventive measures are not yet implemented. Also, they have to underline the importance of the first trimester which is a critical risk period for pregnant women in terms of malaria and anaemia as found in their results.

There are below some comments for the authors that they should take into consideration for improving the manuscript. Moreover, a numbered manuscript would have making easier the review.

1) Suggested title: “Prevalence and risk factors associated with asymptomatic Plasmodium falciparum infection and anaemia among pregnant women at the first antenatal care visit: A hospital based cross-sectional study in Kwale County, Kenya. Use asymptomatic malaria is less specific.

2) Abbreviations: Malaria in pregnancy (MiP and not MIP), Anaemia in pregnancy (AiP and not AIP)

3) Abstract:

a. Background sub-session: Prevalence of malaria in pregnancy, not … in pregnant

b. Results sub-session: please specify what type of odds you used for AiP (crude or adjusted)

4) Key words: important key words do not appear. Please use “asymptomatic malaria” instead of “parasitaemia” which could lead confusion as including symptomatic and asymptomatic cases.

5) Introduction

a. The authors should provide more detail on the current policies against malaria and anaemia during pregnancy used in Kenya, particularly IPTp administration and timeline, iron and folate supplementation? Who in charge of these measures (Government or pregnant women themselves)?

b. Page 3, line 6: “Other factors associated with MiP are……” Use “Other” supposes that you have cited first factors which is not the case. Please rephrase the sentence to make it clear.

c. Page 3, line 7: The authors state that “high parity” is associated with malaria. What do you mean by high parity? Multigravidae women? If yes, I think it’s a wording mistake as it’s well-known that is the primigravidae who are higher risk of MiP, so ‘less parity”.

d. The rational of the study is unclear as presented. Please, give more details explaining what the study brings for the scientific community which are not already known. The only fact that “factors contributing to MiP are not well-described in this part of Kenya” is not sufficient.

e. What is the most common species of malaria parasites in Kenya? I suppose “P. falciparum”. Hence, it would interesting to adjust the title of manuscript accordingly.

6) Methods

a. Ethical statement should be presented in the main text.

i. Did the pregnant women receive IPTp at the 1st ANC if they were eligible?

ii. Did the pregnant women with asymptomatic malaria receive curative treatment? What and how (uncomplicated and severe malaria)? If not, why? Same concerns regarding the anaemia, particularly severe cases?

iii. Any written informed consent? Any ethical committee approval?

b. Study population: give more detail on the strategy of participants selection

c. Did you consider among the symptoms of malaria the history of fever the past 48 h before the visit?

d. Exclusion criteria: Pregnant women who had taken antimalarial drugs within the past two weeks were excluded. What about the women who had taken fever drug? There is a risk to consider women asymptomatic while they just took fever drug the day before the visit.

e. Sample size: The authors have considered a prevalence of MiP from a study in Burkina Faso (24%) while the malaria transmission is different to both countries.

f. Data collection: give more details on socio-demographic characteristics, obstetric and clinical history. How did you assess the LLIN use? Please define the different trimesters of pregnancy (1st, 2nd, 3rd)? Give more details on how the gestational age was assessed?

g. Quality control: Please precise if the 10% of slides chosen was for all sides or positive slides.

h. Statistical analysis: What procedure did you use for variable selection in the final model (multivariate model)?

7) Results

a. Table 1: The proportion of pregnant women in the first trimester at the 1st ANC visit was 9.4% while the authors found that the proportion of pregnant women with gestational age < 16 wg was 15.9%. Why this discrepancy when the first trimester finished at 15 wg.

b. Table 2:

i. Why did you keep in the final multivariate model, the variable gestational age even if not significant in bivariate analysis?

ii. In the same way, why did you keep in the multivariable model “slept under bed net previous night” and “frequency of sleeping under bed net”. Both variables seems to be correlate.

iii. However, you drop out the variable “gravidity” which should be forced in the final model even if not significant because it’s a well-know factors strongly associated with MiP. Furthermore, I would like to suggest to the authors to make an sensitivity analysis by pooling primigravidae and secundigravidae in comparison to multigravidae.

c. Prevalence and factors associated with latent malaria:

i. Please define latent malaria?

ii. What is the prevalence of asymptomatic malaria among pregnant women in the 1st, 2nd and 3rd trimesters at the first ANC visit?

iii. What are the proportion of different species of parasites (P.f.; P.o; P.v; P.m)

iv. Please define POR at the first time it use in the text.

d. Prevalence and factors associated with anaemia:

i. Among the 3.6% of severe anaemia, how many are infected by malaria?

8) Discussion

a. Regarding the factors associated to MiP:

i. The authors should also discuss what happens among women in the first trimester of pregnancy. We can observe that women are more at risk of infection than those in 2nd and 3rd trimester (19.7% vs. 12.7% and 10.4%, respectively).

ii. The only factor associated with MiP is young age (< 20 y). The authors should consider to check an interaction between age and gravidity as both are correlated. Hence, this could be explained by that young pregnant women are mostly primigravidae? This deserves a couple of sentence in the discussion.

b. Regarding the factors associated to AiP: First and 3rd trimester are both associated with AiP. This could be also explained by the haemoglobin level variation due to physiopathology of the pregnancy. This should be included in the discussion

c. Study limitations: The authors have stated several limitations for the study. It is a good point. However, they have to explain how they have controlled this bias to ensure the validity of the study.

9) Conclusion: The authors should revise their conclusion in order to highlight the originality of the study.

6. PLOS authors have the option to publish the peer review history of their article (what does this mean?). If published, this will include your full peer review and any attached files.

Reviewer #1: No

Reviewer #2: No

---

## [Author Response · Author response to Decision Letter 0]

1 Jul 2020

Reviwer #1

Questions/Concerns Comments Corrections 

Major comments

Concern 1 Abstract results: make it clear whether the results presented are adjusted odds ratios throughout. Corrected done it is adjusted prevalence odds ratios that has been used throughout the abstract

Concern 2 Risk factors for AIP: The authors need to better explain how anaemia risk is apparently associated with BOTH first trimester, AND gestation >16 weeks. There are problems with results presentation in Table 3 and it seems they have got the reference and comparator groups switched for the gestation comparison. In addition, part of the results in the table are presented as percentages of women with risk factor who are anaemia (the table column heading) and part are presented as percentage of anaemic women who fall into each risk factor category (not what we want). Please redo this table and recheck your AORs and revise text accordingly. Yes, I have reanalyzed the anemia data, by use of backward elimination method, there seemed an interaction between trimester’s variable and gestational variable. By removing trimester variable for it had a bigger p value, gestational >16 weeks resulted to AOR; 3.3 (1.72-6.41). I have changed the comparison groups accordingly both in the table and in the text

Concern 3 . Discussion: differences in malaria prevalence between Kenya and Burkina Faso are more likely related to malaria transmission intensity, prevalence falls with gestation, and falls more in women receiving IPTp. (Related to this, please also discuss role of IPTp in controlling MIP in conclusions). Interventions measures for first visit pregnant women at antenatal clinic for malaria in pregnancy are; IPTp and provision of LLINs among others [17]. Although the role of IPTp is known to reduce maternal malaria episodes and improve pregnancy outcomes [33], the current study did not include pregnant women who had taken antimalarials hence we could not ascertain the role played by IPTp and majority of our study participants (82.7%) in the current study reported to always using bednet, whereas the study in Burkina Faso included pregnant women seeking ANC services at any point in their pregnancies. This may be factors in the difference between these two reported prevalence. 

Concern 4 Similarly, end P 11, haemodilution over pregnancy seems more likely to explain declining hb over pregnancy. If postulating dfetal needs please provide reference.

 Anemia is a function of plasma volume and red cell mass; both of which increase during pregnancy; but the increase in plasma volume is proportionately greater than the increase in red cell mass [40]. Explanation has been given

Minor comments

Concern 1 Abstract: line 1 “pregnancy”. Results 3rd sentence rewrite “ Women who had MIP were 12.9%”. Last line of results: rewrite for clarity.

 Corrected; pregnant to pregnancy

Rewritten; Women who had plasmodium infections were 12.9 % (40/308)

Concern 2 There were a lot of spacing issues in the pdf, e.g. words joined together, or words and brackets without spaces before them. Examples in abstract methods: odds ratios(OR) and confidence intervals(CI)- there are many others. Attention to details in spacing has been addressed throughout the manuscript 

Concern 3 Introduction 3rd sentence change “malaria with AIP and consumption of soil (geophagy)”, needs different punctuation?

 It has been re-written

Concern 4 . Next sentence: is this risk factors for MIP (parasitic infections? High gravidity is NOT associated with MIP) Or AIP?

 It has been re-written

Concern 5

“In sub-Saharan Africa, MIP affects approximately 125 million pregnant women every year”- this is a major misreading of the cited reference.

 It has been re-written

Concern 6 Top of P 5 “a cross sectional”

 It has been re- written

Concern7

 Inclusion/exclusion: “as well as those who”

 It has been re- written

Concern 8

Concern 8 Sample size “desired level” of what was 5%?

 Two-sided significance level defined at 5%

Concern 9 Data management “ between a variety…”

 It has been re-written 

Concern 10 Page 8: “prevalence and risk factors for malaria parasitemia”. Latent malaria is not a recognised phrase. Next line remove “who”.

 Latent replaced by asymptomatic and re-written 

Concern 11 Parasite counts are best expressed as geometric mean not median. Geometric mean was 3738 

Concern 12 . P 9 defines mild, moderate, severe anaemia.

 Mild, moderate and severe anemia has been defined in definition of terms

Concern 13 Page 11 first paragraph contains multiple grammatical errors, please rewrite

 It has been re looked and corrected accordingly

Concern 14 Third paragraph same page please rewrite. It has been re looked and corrected accordingly

Concern 15 A lot of the references are incomplete (lacking volumes/pages), or not enough details are given to retrieve them, or have formatting issues. Refs 2, 5, 6 (what is MOH? Where can this be obtained- similar comments for some others), 7, 17-22, 33.

The references has been corrected and the links are provided where necessary

Concern 16 Table groups need editing to ensure e.g. it is clear which group women with 8 y school fall into. Same for gestation. And make table 2 N column wider so numbers are readable.

 The table 2; corrected < 8 year in schooling and gestational age in weeks >16

It has been expanded and fonts increased for readability 

Reviwer #2

Comments /questions Corrections/ responses

Concern 1 Suggested title: “Prevalence and risk factors associated with asymptomatic Plasmodium falciparum infection and anaemia among pregnant women at the first antenatal care visit: A hospital based cross-sectional study in Kwale County, Kenya. Use asymptomatic malaria is less specific Plasmodium falciparum was the predominant species more than 80% hence we concur with the reviewer

Prevalence and risk factors associated with asymptomatic Plasmodium falciparum infection and anemia among pregnant women at the first antenatal care visit: A hospital based cross-sectional study in Kwale County, Kenya.

Concern 2 Abbreviations: Malaria in pregnancy (MiP and not MIP), Anaemia in pregnancy (AiP and not AIP) Have corrected the whole manuscript where applicable MIP to MiP and AIP to AiP 

Concern 3 Abstract:

a) Background sub-session: Prevalence of malaria in pregnancy, not … in pregnant Corrected to read Prevalence of malaria in pregnancy, not … in pregnant

 b) Results sub-session: please specify what type of odds you used for AiP (crude or adjusted) We have corrected the omission, the odds Ratio was Adjusted Odd Ratio

 c) Key words: important key words do not appear. Please use “asymptomatic malaria” instead of “parasitaemia” which could lead confusion as including symptomatic and asymptomatic cases Have replaced “Parasitaemia”, with “asymptomatic malaria”

Concern 4 Introduction

a. The authors should provide more detail on the current policies against malaria and anaemia during pregnancy used in Kenya, particularly IPTp administration and timeline, iron and folate supplementation? Who in charge of these measures (Government or pregnant women themselves)?

 “We have included these paragraphs

The focused antenatal care is a package given to pregnant women which entails giving prophylaxis and treatment for anemia and malaria, among other services, where the AiP is prevented by providing nutritional counseling including iron supplements, and treating cases of AiP [20]. 

An intervention aimed at prevention and control MiP adopted by the Ministry of Health (MOH) Kenya, are; Intermittent Preventive Treatment (IPTp) of MiP, with Sulfadoxine Pyrimethamine (SP) given after 12 weeks gestational period done four weeks apart until the pregnant woman delivers, in areas with high malaria transmission such as Western, Nyanza and Coast regions and Long-lasting Insecticidal Nets (LLINs) at the ANC in the first contact among others [21]

 Page 3, line 6: “Other factors associated with MiP are……” Use “Other” supposes that you have cited first factors which are not the case. Please rephrase the sentence to make it clear. Besides plasmodium infections other factors known to cause AiP include nutritional deficiencies, infectious diseases like HIV, parasitic infections like hookworm infestation, and the hemoglobinopathies [4, 5].

 c. Page 3, line 7: The authors state that “high parity” is associated with malaria. What do

 you mean by high parity? Multigravidae women? If yes, I think it’s a wording mistake as it’s well-known that is the primigravidae who are higher risk

of MiP, so ‘less parity”.

 Corrected; Several studies has documented factors associated with MiP are women with less parity,

 d. The rational of the study is unclear as presented. Please, give more details explaining what the study brings for the scientific community which are not already known. The only fact that “factors contributing to MiP are not well-described in this part of Kenya” is not sufficient.

 Kenya Malaria Indicator Survey (KMIS) 2015 reported 2 folds increase in the prevalence of malaria in the coastal region compared to KMIS 2010 [18]. There are a possibility plasmodium infections in healthy adults, including pregnant women, in moderate to high transmission areas rarely result in fever [19]. Therefore, elimination of malaria is highly unlikely if diagnostic strategies do not include asymptomatic patients, because they will remain a reservoir of parasites contributing to the spread of the disease from one malaria season to the next. Man¬agement and control of MiP and AiP are enhanced by the availability of local prevalence statistics, which is not adequately provided in Kwale County except the data collected in passive surveillance. It is in this background we carried out this study to estimate the prevalence of asymptomatic MiP and AiP and identify the associated factors among women attending their first ANC visit at the largest referral health facility in Kwale County.

 e. What is the most common species of malaria parasites in Kenya? I suppose “P. falciparum”. Hence, it would interest to adjust the title of manuscript accordingly.

 Has adjusted the title by including the “asymptomatic Plasmodium falciparum infection”

Concern 5 Methods

a) Ethical statement should be presented in the main text.

i) Did the pregnant women receive IPTp at the 1st ANC if they were eligible?

 Yes, they did, as per the government of Kenya Policy in prevention and control of Malaria in Pregnancy. .http://www.nmcp.or.ke/index.php/malaria-in-pregnancy

ii. Did the pregnant women with asymptomatic malaria receive curative treatment? What and how (uncomplicated and severe malaria)? If not, why? Same concerns regarding the anaemia, particularly severe cases? Following informed consent, the study participants were explained whose test positive for malaria and will have anemia will benefit by being treated as per the guidelines of malaria and anemia in pregnancy with no extra cost. 

iii. Any written informed consent? Any ethical committee approval?

 We sought written, informed consent from each participant before interviewing and finger pricking for malaria blood slides and Hemoglobin level analysis. Permission was granted to conduct the study by Kwale Department of Health and Msambweni Hospital Director. Ethical clearance was obtained from the Pwani University -Ethical Review Committee (ERC/MSc/021/2018).

 b. Study population: give more detail on the strategy of participant’s selection

 Sampling procedures

Systematic random sampling method was used to select study participants. Our sampling interval was based on the daily entries in the mother-child health (MCH) register from September 2018 to February 2019. The sampling started by selecting a participant from the daily entries list at random using a table of random numbers and then every kth participant in the frame was selected. A selection interval (k) was determined by dividing the total daily entry listed in order to get the number of the participants required per day. If a randomly-selected participant was not eligible for an interview or refused to be part of the study, the next eligible participant on the list was selected. We sampled our study participants until we arrived at our desired sample size of 308. 

Following informed consent, the study participants were explained whose test positive for malaria and anemia will benefit by being treated as per the guidelines of malaria and anemia in pregnancy with no extra cost. 

 c. Did you consider among the symptoms of malaria the history of fever the past 48 h before the visit?

Exclusion criteria: Pregnant women who had taken antimalarial drugs within the past two weeks were excluded. What about the women who had taken fever drug? There is a risk to consider women asymptomatic while they just took fever drug the day before the visit. Asymptomatic malaria was defined as the presence in the peripheral blood of asexual blood stage of Plasmodium, irrespective of species but has no symptoms of malaria per clinical assessment (i.e. temperature <37.50C, chills, rigor, nausea, vomiting, headache, anorexia, or joint/muscle pains) and has not taken antipyretics within 48 hours and antimalarials within 14 days. 

Concern 6 e. Sample size: The authors have considered a prevalence of MiP from a study in Burkina Faso (24%) while the malaria transmission is different to both countries. 

 True, we took the prevalence of MiP for Burkina Faso (24%). To our best of our knowledge by the time we were conceptualizing the protocol there was no study for asymptomatic MiP we could get hence we used the African study similar to ours to calculate the prevalence. Also, we thought the differences in malaria prevalence between Kenya and Burkina Faso are more likely related to malaria transmission intensity due to we have similar climate is characterized by ‘long rains’ (April–June) and ‘short rains’ (October–December) rainy seasons.

 f. Data collection: give more details on socio-demographic characteristics, obstetric and clinical history. 

 Variables collected included; 

Socio-demographic characteristics; mother's age, education level, marital status and occupation.

Obstetrics variables; gravidity, parity, trimesters, and gestational age in weeks.

Clinical history variables; history of fever in the last 48 hours and taken anti pyretic drugs, whether the client has taken antimalarials drugs, tendency of geophagy, whether patient is on iron supplements 

How did you assess the LLIN use? 

 We depended on what the study participant reported which has been highlighted as a limitation but we used a trained nurse on the protocol to collect the data.

 Please define the different trimesters of pregnancy (1st, 2nd, 3rd)? 

 First trimester was defined as; from week 1 to the end of week 12 while the second trimester is from week 13 to the end of week 26 and the third trimester is from week 27 to the end of the pregnancy

 Give more details on how the gestational age was assessed? We used fundal height and last monthly period to estimate the gestational age

Concern 7 g. Quality control: Please precise if the 10% of slides chosen was for all sides or positive slides. An independent qualified parasitologist examined 10% of both positives and negatives slides which were randomly selected

Concern 8 h. Statistical analysis: What procedure did you use for variable selection in the final model (multivariate model)?

 Variables with p-value ≤ 0.20 were included in a logistic regression model using a backward stepwise elimination method to identify independently associated factors.

Concern 9 

7) Results

a. Table 1: The proportion of pregnant women in the first trimester at the 1st ANC visit was 9.4% while the authors found that the proportion of pregnant women with gestational age < 16 wg was 15.9%. Why this discrepancy when the first trimester finished at 15 wg.

b. Table 2:

i. Why did you keep in the final multivariate model, the variable gestational age even if not significant in bivariate analysis?

ii. In the same way, why did you keep in the multivariable model “slept under bed net previous night” and “frequency of sleeping under bed net”. Both variables seems to be correlate.

iii. However, you drop out the variable “gravidity” which should be forced in the final model even if not significant because it’s a well-know factors strongly associated with MiP. Furthermore, I would like to suggest to the authors to make an sensitivity analysis by pooling primigravidae and secundigravidae in comparison to multigravidae.

c. Prevalence and factors associated with latent malaria:

i. Please define latent malaria?

ii. What is the prevalence of asymptomatic malaria among pregnant women in the 1st, 2nd and 3rd trimesters at the first ANC visit?

iii. What are the proportion of different species of parasites (P.f.; P.o; P.v; P.m)

iv. Please define POR at the first time it use in the text.

d. Prevalence and factors associated with anaemia:

i. Among the 3.6% of severe anaemia, how many

a) The difference is due to the definition of the trimesters that was used in this study. First trimester was defined as; from week 1 to the end of week 12 while the second trimester is from week 13 to the end of week 26 and the third trimester is from week 27 to the end of the pregnancy

b) Have reanalyzed the data (see table 2 and 3)

c) Similar to the answer b, have re- analyzed the data (Table 2) and pooled together the primigravidae and second gravidae in comparison with multigravidae but still was not significant

we have used I have replaced latent malaria with asymptomatic malaria 

Gestational trimesters with plasmodium infections, the first trimester were 2/29 (6.9%), second trimester 24/173 (13.9%) and third trimester were 14/106 (13.2%). 

 Plasmodium falciparum were 35 (87.5%), Plasmodium malarie 3 (7.5%) and Plasmodium ovale 2 (5.0%).

I has been defined as Prevalence Odds Ratio

Those who had severe anemia 4/7 (57.1%) had malaria, moderate anemia 15/95 (15.8%) and mild anemia was 14/90 (15.6%).

 Discussion

a. Regarding the factors associated to MiP:

i. The authors should also discuss what happens among women in the first trimester of pregnancy. We can observe that women are more at risk of infection than those in 2nd and 3rd trimester (19.7% vs. 12.7% and 10.4%, respectively).

 The current study reported a higher proportion of Plasmodium infections in pregnant women who were both in second and third trimesters and less proportion to the first trimester. Studies in Nigeria have reported high malaria prevalence in pregnant women who were in their second trimesters [27-29]. A study in Mali reported pregnant women in their first trimester were two times more likely to get malaria compared to the third trimester [30]. In contrast, our study reported less proportion of pregnant women in their first trimester and was not associated with malaria. This was probably due to the small number of pregnant women among this category. With an increase in the number of pregnant women in their first trimester, there is the possibility that there could be changes from the present results. 

 ii. The only factor associated with MiP is young age (< 20 y). The authors should consider to check an interaction between age and gravidity as both are correlated. Hence, this could be explained by that young pregnant women are mostly primigravidae? This deserves a couple of sentence in the discussion. 

There was no Interaction 

In this study, the highest proportional of Plasmodium infections was observed among the primigravidae (19.7%): followed by secundigravidae (12.7.7%) and multigravidae (10.4 %) with parasitaemia declining with increasing gravidity. These results are consistent with previous reports which found plasmodium infections are more common in primigravidae women compared to multigravidae women [7, 28, 31]. The reason for the present result of gravidae-associated predisposition to P. falciparum infections may be due to the fact that adults who live in malaria-endemic regions generally have some acquired immunity to malaria infection due to immunoglobulin production stimulated by previous malaria infection. This acquired immunity diminishes significantly in pregnancy particularly in primigravidae. It has also been suggested by various authors that the early onset of antibody response in multigravidae and the delayed antibody production in primigravidae may be responsible for the gravidity-dependent and differential prevalence of falciparum malaria among pregnant women [19, 32].

 b. Regarding the factors associated to AiP: First and 3rd trimester are both associated with AiP. This could be also explained by the haemoglobin level variation due to physiopathology of the pregnancy. This should be included in the discussion

 We found that anemia is more common among women in their third trimester than among women in their first trimester, similar to findings reported in other studies [39, 40]. Hemoglobin decreases until the end of the third-trimester. This might be due the fact that increase in trimester may cause reduction in maternal iron reserves. Anemia is a function of plasma volume and red cell mass; both of which increase during pregnancy; but the increase in plasma volume is proportionately greater than the increase in red cell mass [41]. Also, anemia in the third trimester may be more likely due to higher nutrient demands of the fetus later in pregnancy

c. Study limitations: The authors have stated several limitations for the study. It is a good point. However, they have to explain how they have controlled this bias to ensure the validity of the study.

 We collected data from September 2018 to February 2019, a period during which there is low malaria transmission. This could have resulted in the underestimation of the overall prevalence of asymptomatic MiP in our study area. A continuous monitoring throughout the year of MiP incidences will account for seasonality burden [21].

In addition, the study was hospital-based, excluding pregnant women who did not seek ANC services. While this may limit the generalizability of findings to the community, few women fail to seek antenatal care in our study area. Determination of factors associated with asymptomatic MiP and AiP in hospital based studies provides a proxy indicator of predictors in the community of that particular facility when community based surveys are not feasible. 

 Lastly, this study did not explore other factors that may contribute to anemia, including nutritional factors, soil-transmitted helminthes infection, and hereditary conditions such as sickle cell disease thus limiting our ability to assess the contribution of other causes of anemia during pregnancy. However, diagnosis of anemia was based on laboratory analysis and did not depend on clinical assessment as reported by other researchers.

Concern 10 9) Conclusion: The authors should revise their conclusion in order to highlight the originality of the study.

Asymptomatic Plasmodium infections and anemia are common in women attending their first ANC visit at Msambweni County Referral Hospital in Kwale County. Most of the Plasmodium infections in this area are caused by P. falciparum. Asymptomatic MiP was associated with younger maternal age (≤20 years). Anemia in pregnancy was associated with Plasmodium infections, women who reported to have geophagy tendency and those who were their third trimester. In the study area, we recommend pregnant women should not delay their first ANC attendance, for less than 10% attended in their first trimester. All women of childbearing age should be included in measures to control Plasmodium infection and anemia by the National Malaria Control Program, reproductive health department and other non – state actors should. Also, the reproductive health department should carry out health promotion and education on late adolescent and school going pregnancy for delay of sexual debut.

---

## [Decision Letter · Decision Letter 1]

28 Jul 2020

PONE-D-20-11384R1

Prevalence and risk factors associated with asymptomatic Plasmodium falciparum infection and anemia among pregnant women at the first antenatal care visit: A hospital based cross-sectional study in Kwale County, Kenya.

PLOS ONE

Dear Dr. Nyamu,

Thank you for submitting your revised manuscript to PLOS ONE. After careful consideration, we feel that it has merit but does not fully meet PLOS ONE’s publication criteria as it currently stands. Therefore, we invite you to submit a newly revised version of the manuscript that addresses the points raised by the reviewers.

Both reviewers were in agreement that the revised manuscript is an improvement on the original submission, but both have nevertheless still raised some issues that require further attention, in particular concerning grammatical errors. Reviewer #1 has taken the time to provide a copy-edited version, appended to this letter, to assist in improving the English throughout. You should address these concerns in your revision along with the specific remaining issues outlined in the appended comments from the reviewers.

We look forward to receiving your revised manuscript.

Kind regards,

Adrian J.F. Luty, PhD

Academic Editor

PLOS ONE

Reviewers' comments:

Reviewer's Responses to Questions

**Comments to the Author**

1. If the authors have adequately addressed your comments raised in a previous round of review and you feel that this manuscript is now acceptable for publication, you may indicate that here to bypass the “Comments to the Author” section, enter your conflict of interest statement in the “Confidential to Editor” section, and submit your "Accept" recommendation.

Reviewer #1: (No Response)

Reviewer #2: All comments have been addressed

2. Is the manuscript technically sound, and do the data support the conclusions?

Reviewer #1: Partly

Reviewer #2: Partly

3. Has the statistical analysis been performed appropriately and rigorously? 

Reviewer #1: Yes

Reviewer #2: Yes

4. Have the authors made all data underlying the findings in their manuscript fully available?

Reviewer #1: Yes

Reviewer #2: Yes

5. Is the manuscript presented in an intelligible fashion and written in standard English?

Reviewer #1: No

Reviewer #2: No

6. Review Comments to the Author

Reviewer #1: The authors have addressed many of my comments but I have a few of remaining issues they need to address.

It would have been helpful if the manuscript had page and line numbers.

1. Discussion page 3: The high rates of net ownership cannot really be attributed to the policy of bed net distribution at first ANC visit as women would only receive these nets AFTER enrolment in the study- so no primigravidae would have received nets by this route. The mass distribution campaigns are more likely driving this.

2. Table 3 still needs correction. The column n(%) needs to be the percentage of women who are anaemic as a fraction of the women in that row. From Trimester onwards the figures must be corrected.

3. In the same table the crude POR for secundigravidae is incorrect. They have a higher anaemia prevalence than women in first trimester.

4. Also in Table 3, please switch “gestation >16 weeks” and “gestation ≤ 16 weeks” to be consistent with the rest of the table where lower values come first.

5. In these lines there is an error in n for women ≥16 weeks with anaemia- is it 174?

I have also attached the revised manuscript with highlighting to indicate the many minor formatting, spacing and other editorial issues that need to be corrected. These are mostly around missing spaces, and spaces which should not be there.

Two specific formatting comments:

1. Primigravidae, multigravidae are always written as one word and are nouns. Primi-gravidae is not correct and nor is “primigravidae women.

2. A semi-colon has a different function from the usage here. Colons are appropriate, or the punctuation can be removed.

Reviewer #2: While the revised manuscript has improved, it still requires significant grammatical editing to improve its readability and clarity. Some concepts remain unaddressed for me.

a) In the first review, I have highlighted that the rational of the study is unclear. The authors stated that pregnant women are potential reservoir of P.f parasites; however, the IPTp is one the main strategy protected pregnant women and is administered regardless the malaria status of women. Hence, this should not be a major lack as the women will clean by IPTp. In addition, in their conclusion, the authors recommended to take into account women of reproductive age in malaria control program; but no specific strategy has been suggested. The authors should revise their manuscript to show the scientific add-value of their paper.

b) For positive malaria management, the authors just precised that they used per the government of Kenya

Policy in prevention and control of Malaria in Pregnancy. But they should presented shortly how these cases are managed in the manuscript.

c) Informed consent administration: What is the process for illiterate women who did not write?

d) There is no flow chart diagram? How many pregnant women ineligible were excluded to reach the 308 pregnant women? And what are the reasons? Also compared the baseline characteristics of the pregnant women included and those who did not

e) How many participants have been excluded because of the following criteria: Intake of antipyretic drugs within 48h?

f) I have a concerns with the cut-off used to define 1st, 2nd and 3rd trimester. The authors define first trimester by using a gestational age < 12 wg instead of 14 completed wg, the cut-off usually used. Why ?

g) Two methods were used to measure the gestational age (fundal height and LMP); but there is no precision in what situation one or both are used?

h) QC of positive and negative slides. What is done in case of discrepancy ?

i) The authors showed that the prevalence of malaria were 6.9%; 13.9% and 13.2% in the first, 2nd and 3rd trimester respectively? It is slightly amazing as results as the prevalence should decrease from first trimester to delivery because of preventive measures implemented. How the authors explained this result. this deserve a couple of sentences in the discussion session

j) The authors highlighted the IPTp started from the 12 wg. I'm not sure if this is the current WHO recommendation as the IPTp should started as soon as possible from the 2nd trimester.

k) Number the page of manuscript.

7. PLOS authors have the option to publish the peer review history of their article (what does this mean?). If published, this will include your full peer review and any attached files.

Reviewer #1: No

Reviewer #2: **Yes: **Manfred Accrombessi

---

## [Author Response · Author response to Decision Letter 1]

17 Aug 2020

Thanks for the consideration to review and your reviwers comments have been addressed to our best of knowledge.

---

## [Decision Letter · Decision Letter 2]

10 Sep 2020

Prevalence and risk factors associated with asymptomatic Plasmodium falciparum infection and anemia among pregnant women at the first antenatal care visit: A hospital based cross-sectional study in Kwale County, Kenya.

PONE-D-20-11384R2

Dear Dr. Nyamu,

We’re pleased to inform you that your manuscript has been judged scientifically suitable for publication and will be formally accepted for publication once it meets all outstanding technical requirements.

Kind regards,

Adrian J.F. Luty, PhD

Academic Editor

PLOS ONE

Additional Editor Comments (optional):

This manuscript is now acceptable for publication; however, the English needs again additional improvement.

For example on the clean version, there are some minor errors identified

- Line 1: Plasmodium should be in italic

- Line 2: Typo error on "department"

- Line 26: "Prevalence of malaria in pregnancy" instead of "both asymptomatic and symptomatic"

- Line 30: delete "asymptomatic". Pregnant women could not be asymptomatic

- Line 33: Odds ==> odds

- Line 44: Asymptomatic MiP, not AiP

- Line 46: Please reconsider "Pregnant"; may be "Pregnancy" or "Pregnant women"

- Line 54: Sub - Saharan Africa ==> sub-Saharan Africa

- Line 55: Delete 89324/142896 and keep only the %

- Line 63: Malaria in pregnancy, not Pregnancy with capital P. And we have already defined an abbreviations for that purpose, so use MiP

- Lines 65-67: Rephrase the sentence. Due to...., pregnant women are more at risk of malaria"

-Lines 70-71: Rephrase the sentence to make it clear. What do you mean by focused antenatal care? Use abbreviation of ANC previously defined

- Lines 105, 108: Combine sub-session of study design and study population ==> study design and population

- Lines 111: Inclusion and exclusion criteria should be included in the "study population" session

- Line 128-136: Rephrase the sub-session "sample size determination" by making an unique block without bullets.

- Line 156: Delete were, there is a repetition of "were" with those of line 155

- Line 174: Why abbreviate No Parasit Found (NPF) as you did not use it after in the mansucript

- Line 217: First bracket missing for 56.2%

Reviewers' comments:

Reviewer's Responses to Questions

**Comments to the Author**

1. If the authors have adequately addressed your comments raised in a previous round of review and you feel that this manuscript is now acceptable for publication, you may indicate that here to bypass the “Comments to the Author” section, enter your conflict of interest statement in the “Confidential to Editor” section, and submit your "Accept" recommendation.

Reviewer #1: All comments have been addressed

Reviewer #2: All comments have been addressed

2. Is the manuscript technically sound, and do the data support the conclusions?

Reviewer #1: (No Response)

Reviewer #2: Yes

3. Has the statistical analysis been performed appropriately and rigorously? 

Reviewer #1: (No Response)

Reviewer #2: Yes

4. Have the authors made all data underlying the findings in their manuscript fully available?

Reviewer #1: (No Response)

Reviewer #2: Yes

5. Is the manuscript presented in an intelligible fashion and written in standard English?

Reviewer #1: (No Response)

Reviewer #2: No

6. Review Comments to the Author

Reviewer #1: (No Response)

Reviewer #2: (No Response)

7. PLOS authors have the option to publish the peer review history of their article (what does this mean?). If published, this will include your full peer review and any attached files.

Reviewer #1: No

Reviewer #2: No

---

## [Editor Report · Acceptance letter]

25 Sep 2020

PONE-D-20-11384R2 

Prevalence and risk factors associated with asymptomatic *Plasmodium falciparum* infection and anemia among pregnant women at the first antenatal care visit: A hospital based cross-sectional study in Kwale County, Kenya. 

Dear Dr. Nyamu:

I'm pleased to inform you that your manuscript has been deemed suitable for publication in PLOS ONE. Congratulations! Your manuscript is now with our production department. 

Kind regards, 

on behalf of

Dr. Adrian J.F. Luty 

Academic Editor

PLOS ONE